# Sharp Calibrated Gaussian Processes

**Alexandre Capone**
Technical University of Munich
alexandre.capone@tum.de

**Sandra Hirche**
Technical University of Munich
hirc@cit.tum.de

**Geoff Pleiss**
University of British Columbia
Vector Institute
geoff.pleiss@stat.ubc.ca

## Abstract

While Gaussian processes are a mainstay for various engineering and scientific applications, the uncertainty estimates don't satisfy frequentist guarantees and can be miscalibrated in practice. State-of-the-art approaches for designing calibrated models rely on inflating the Gaussian process posterior variance, which yields confidence intervals that are potentially too coarse. To remedy this, we present a calibration approach that generates predictive quantiles using a computation inspired by the vanilla Gaussian process posterior variance but using a different set of hyperparameters chosen to satisfy an empirical calibration constraint. This results in a calibration approach that is considerably more flexible than existing approaches, which we optimize to yield tight predictive quantiles. Our approach is shown to yield a calibrated model under reasonable assumptions. Furthermore, it outperforms existing approaches in sharpness when employed for calibrated regression.

## 1 Introduction

Gaussian process (GP) regression offers an ambitious proposition: by conditioning a model on measurement data, we are provided with a Gaussian probability distribution for the unseen data. Assuming that the posterior probability distribution holds, we can then directly calibrate our model using the inverse error function. Though the distribution of unseen data seldom follows the Gaussian prior distribution, and the GP generally does not adapt adequately to the observed distributions after being conditioned on the data, GPs have become one of the most powerful and established regression techniques. Besides having found widespread use in machine learning (Deisenroth et al., 2015; Srinivas et al., 2012), their good generalization properties have motivated applications in the fields of control (Kocijan, 2016), astrophysics (Roberts et al., 2013) and chemistry (Deringer et al., 2021), to name a few. Furthermore, the Bayesian paradigm offers a powerful tool to analyze the theoretical properties of different regression techniques (Srinivas et al., 2012; Capone et al., 2022).

In this paper, we present a novel approach to obtaining sharp calibrated Gaussian processes, i.e., Gaussian processes that provide concentrated predictive distributions that accurately match the observed data. Instead of computing confidence intervals by inflating the Gaussian process posterior variance, our approach discards it and computes a new quantity inspired by the computation of the posterior variance, where all hyperparameters are chosen in a way that results in both accurate and sharp calibration. In other words, we train two separate Gaussian processes: one for the predictive mean and one for obtaining predictive quantiles, which is exclusively used for calibration purposes. By doing so, we reach considerably more flexibility than existing calibration approaches, which enables us to additionally optimize the sharpness of the calibration. Our approach outperforms several

37th Conference on Neural Information Processing Systems (NeurIPS 2023).

state-of-the-art calibration approaches in terms of sharpness while still yielding similar calibration performance. Furthermore, it is competitive compared to a neural network-based method in sharpness without sacrificing calibration performance.

**Notation.** We use $\mathbb{R}_+$ to denote the non-negative real numbers. Boldface lowercase/uppercase characters denote vectors/matrices. For two vectors $\boldsymbol{a}$ and $\boldsymbol{a}'$ in $\mathbb{R}^d$, we employ the notation $\boldsymbol{a} \leq \boldsymbol{a}'$ to denote componentwise inequality, i.e., $a_i \leq a_i'$, $i = 1, \ldots, d$. For a square matrix $\boldsymbol{K}$, we use $|\boldsymbol{K}|$ to denote its determinant, and $[\boldsymbol{K}]_{ij}$ to denote the entry corresponding to the $i$-th row and $j$-th column.

# 2 Related Work

**Calibration of Classification Models.** There has been extensive work on obtaining calibrated models in the domain of classification. While there are many methods that do not employ post-processing, we only focus here on methods that employ some form of post-processing. Most forms of post-processing-based calibration for classification fall into the category of conformal methods (Vovk et al., 2005), which, given an input, aim to produce sets of labels that contain the true label with a pre-specified probability. Arguably the two most common forms of calibration are isotonic regression (Niculescu-Mizil & Caruana, 2005) and Platt scaling (Platt et al., 1999). In Niculescu-Mizil & Caruana (2005), Platt scaling and isotonic regression are analyzed extensively for different types of predictive models. In Guo et al. (2017), a modified form of Platt scaling for modern classification neural networks is proposed.

**Calibration of Regression Models.** Though initially developed for classification, conformal calibration has been extended to regression settings. In Lakshminarayanan et al. (2017), a calibration approach was proposed for deep ensembles. Gal et al. (2017) propose a dropout-based technique for calibrating deep neural networks. However, these approaches require changing the regressor, potentially deteriorating its predictive performance. It should be noted that Bayesian neural networks (MacKay, 1995), while being able to provide credible sets for the output, fully trust the posterior, resulting in a naively calibrated model that seldom reflects the data's distribution. As a remedy for this, Kuleshov et al. (2018) present a recalibration approach that scales a model's predictive quantiles to satisfy the observed data's distribution. In Vovk et al. (2020), a similar approach is presented, where interpolation between scaling factors is randomized, and a theoretical analysis is provided. An extension of both Kuleshov et al. (2018) and Vovk et al. (2020) and other recalibration is proposed in Marx et al. (2022), along with corresponding theoretical guarantees. While these methods have been shown to yield well-calibrated models, the resulting predictive quantiles are potentially much too crude, resulting in predictions that perform poorly in terms of sharpness, i.e., the corresponding confidence intervals will overestimate the model error by a very large margin. To remedy this, Song et al. (2019) and Kuleshov & Deshpande (2022) propose optimizing the parameters of a recalibration model by obtaining calibration on a distribution level. However, while Song et al. (2019) relies on complex approximations and provides no theoretical guarantees, Kuleshov & Deshpande (2022) does not allow to optimize for sharpness directly, and calibration is only guaranteed asymptotically as the number of data grows.

# 3 Problem Statement

Consider a compact input space $\mathcal{X} \subset \mathbb{R}^d$, and output space $\mathcal{Y} \subset \mathbb{R}$, and an unknown data distribution $\Pi$ on $\mathcal{X} \times \mathcal{Y}$. Consider a model conditioned on training data $\mathcal{D}_{\mathrm{tr}} \sim \Pi$ that, for every $\boldsymbol{x} \in \mathcal{X}$ and confidence level $\delta \in [0, 1]$, returns a base prediction $\mu_{\mathcal{D}_{\mathrm{tr}}}(\boldsymbol{x})$ and an additive cut-point term $\beta_\delta \sigma_{\mathcal{D}_{\mathrm{tr}}}(\delta, \boldsymbol{x})$, where $\sigma_{\mathcal{D}_{\mathrm{tr}}}(\delta, \boldsymbol{x}) \geq 0$ and $\beta_\delta \in \mathbb{R}$ is potentially negative, such that $\mu_{\mathcal{D}_{\mathrm{tr}}}(\boldsymbol{x}) + \beta_\delta \sigma_{\mathcal{D}_{\mathrm{tr}}}(\delta, \boldsymbol{x})$ corresponds to a predictive $\delta$-quantile. The model is then said to be calibrated if

$$\mathbb{P}_{\boldsymbol{x}, y \sim \Pi} \Big( y - \mu_{\mathcal{D}_{\mathrm{tr}}}(\boldsymbol{x}) \leq \beta_\delta \sigma_{\mathcal{D}}(\delta, \boldsymbol{x}) \Big) = \delta \tag{1}$$

holds for every $\delta \in [0, 1]$. Furthermore, the calibrated model is also said to be sharp if the corresponding predictive distributions are concentrated (Gneiting et al., 2007), i.e., if the centered confidence intervals induced by the predictive quantiles

$$|\beta_\delta \sigma_{\mathcal{D}_{\mathrm{tr}}}(\delta, \boldsymbol{x}) - \beta_{1-\delta} \sigma_{\mathcal{D}_{\mathrm{tr}}}(1 - \delta, \boldsymbol{x})| \tag{2}$$

are as small as possible for every $\delta \in [0, 1]$. Our goal is to find a sharply calibrated model based on GP regression.

## 4 Gaussian Process Regression

In this section, we briefly review GP regression, with particular focus on the choice and influence of hyperparameters.

A GP is formally defined as a collection of random variables, any finite subset of which is jointly Gaussian (Rasmussen & Williams, 2006). It is fully specified by a prior mean function, which we set to zero without loss of generality, and a hyperparameter-dependent covariance function, called kernel $k : \boldsymbol{\Theta} \times \mathcal{X} \times \mathcal{X} \to \mathbb{R}$, where $\boldsymbol{\Theta}$ denotes the hyperparameter space. The core concept behind GP regression lies in assuming that any finite number of measurements of an unknown function $f : \mathcal{X} \to \mathbb{R}$ at arbitrary inputs $\boldsymbol{x}_1, \ldots, \boldsymbol{x}_N \in \mathcal{X}$ are jointly Gaussian with mean zero and covariance $\boldsymbol{K}(\boldsymbol{\theta})$, where

$$[\boldsymbol{K}(\boldsymbol{\theta})]_{ij} = k(\boldsymbol{\theta}, \boldsymbol{x}_i, \boldsymbol{x}_j)$$

consists of kernel evaluations at the test inputs.

Given a set of noisy observations $\mathcal{D} = \{\boldsymbol{x}_i, y_i\}_{i=1}^N$, where $y_i := f(\boldsymbol{x}_i) + \varepsilon_i$ and $\varepsilon_i \sim \mathcal{N}(0, \sigma_0^2)$ is iid Gaussian measurement noise, we can condition the GP on them to obtain the posterior distribution. The posterior distribution for a new observation $y^*$ at an arbitrary input $\boldsymbol{x}^*$ is again Gaussian distributed, with mean and variance

$$\mu_{\mathcal{D}_{\mathrm{tr}}}(\boldsymbol{\theta}, \boldsymbol{x}^*) = \boldsymbol{k}(\boldsymbol{\theta}) \left( \boldsymbol{K}(\boldsymbol{\theta}) + \sigma_0^2 \boldsymbol{I} \right)^{-1} \boldsymbol{y}, \tag{3a}$$

$$\sigma_{\mathcal{D}_{\mathrm{tr}}}^2(\boldsymbol{\theta}, \boldsymbol{x}^*) = k(\boldsymbol{\theta}, \boldsymbol{x}^*, \boldsymbol{x}^*) - \boldsymbol{k}(\boldsymbol{\theta}) \left( \boldsymbol{K}(\boldsymbol{\theta}) + \sigma_0^2 \boldsymbol{I} \right)^{-1} \boldsymbol{k}(\boldsymbol{\theta}) + \sigma_0^2, \tag{3b}$$

with $\boldsymbol{k}(\boldsymbol{\theta}) := (k(\boldsymbol{\theta}, \boldsymbol{x}^*, \boldsymbol{x}_1), \ldots, k(\boldsymbol{\theta}, \boldsymbol{x}^*, \boldsymbol{x}_N))$, $\boldsymbol{y} = (y_1, \ldots, y_N)$, and $\boldsymbol{I}$ denoting the identity matrix. In this paper, we restrict ourselves to kernels $k(\cdot, \cdot, \cdot)$ that yield a posterior variance that is monotonically increasing with respect to the hyperparameters $\boldsymbol{\theta}$, as specified in the following assumption.

**Assumption 4.1.** The posterior variance $\sigma_{\mathcal{D}_{\mathrm{tr}}}^2(\boldsymbol{\theta}, \boldsymbol{x}^*)$ is a continuous function of $\boldsymbol{\theta}$. Furthermore, for all hyperparameters $\boldsymbol{\theta}, \boldsymbol{\theta}' \in \boldsymbol{\Theta}$ with $\boldsymbol{\theta} \leq \boldsymbol{\theta}'$, it holds that $\sigma_{\mathcal{D}_{\mathrm{tr}}}^2(\boldsymbol{\theta}, \boldsymbol{x}^*) \leq \sigma_{\mathcal{D}_{\mathrm{tr}}}^2(\boldsymbol{\theta}', \boldsymbol{x}^*)$.

Assumption 4.1 holds trivially for the signal variance of a kernel. Moreover, it holds for any hyperparameters that lead to a monotonous increase in the Fourier transform, which is the case, e.g., for the inverse lengtschale of stationary kernels up to a multiplicative factor corresponding to the ratio of the lengthscales (Capone et al., 2022). Furthermore, several results that employ the so-called fill-distance indicate that Assumption 4.1 holds for the inverse lengthscale of a broad class of stationary kernels as the lengthscale becomes very large or very small (Wendland, 2004). In our experiments, we observed that Assumption 4.1 was never violated for the inverse lengthscale of the squared-exponential kernel. Assumption 4.1 will be leveraged to define a cumulative density function by also changing the hyperparameters corresponding to the posterior covariance, as opposed to simply scaling it.

Arguably one of the most challenging aspects of GP regression lies in the choice of hyperparameters $\boldsymbol{\theta}$, as they ultimately determine various characteristics of the posterior, e.g., smoothness and amplitude. In practice, the most common way of choosing $\boldsymbol{\theta}$ is by maximizing the log marginal likelihood

$$\log p(\boldsymbol{y}|\boldsymbol{X}, \boldsymbol{\theta}) = -\frac{1}{2} \log |\boldsymbol{K}(\boldsymbol{\theta}) + \sigma_0^2 \boldsymbol{I}| - \frac{N}{2} \log(2\pi) - \frac{1}{2} \boldsymbol{y}^\mathsf{T} \left( \boldsymbol{K}(\boldsymbol{\theta}) + \sigma_0^2 \boldsymbol{I} \right)^{-1} \boldsymbol{y}. \tag{4}$$

In terms of *posterior mean* quality, i.e., predictive performance of $\mu_{\mathcal{D}_{\mathrm{tr}}}(\boldsymbol{\theta}, \boldsymbol{x}^*)$, choosing the hyperparameters in this manner is often the most promising option, since it seeks a trade-off between model complexity and data fit, and has repeatedly been shown to yield a satisfactory mean square error when applied to test data (Rasmussen & Williams, 2006). However, even when employing log-likelihood maximization, the posterior GP distribution is seldom well calibrated, and, in practice, the data often has a significantly different distribution. As a result, quantiles obtained with a purely Bayesian approach are either too confident or too conservative in practice (Capone et al., 2022; Fong & Holmes, 2020). Furthermore, the restrictions imposed by the GP prior often produce a posterior variance $\sigma^2(\boldsymbol{\theta}, \boldsymbol{x}^*)$ that is grossly conservative. See Figure 1 for an illustration.

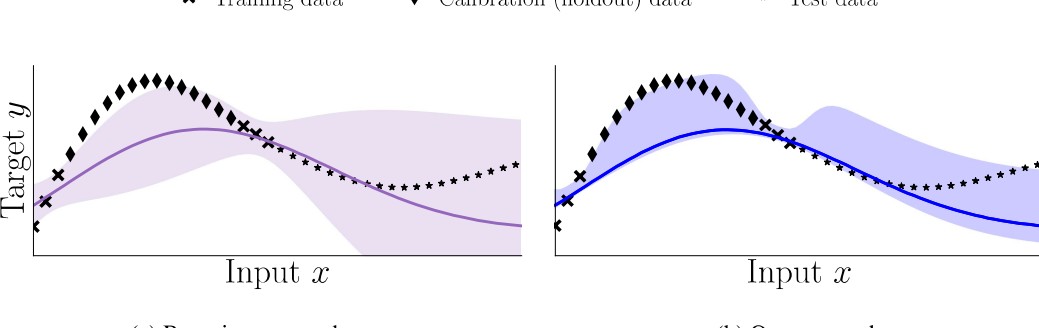

(a) Bayesian approach.          (b) Our approach.

Figure 1: Confidence interval of $95\%$ (shaded regions) obtained with purely Bayesian approach (a), where the inverse error function is employed to compute $\beta_\delta$, and our approach (b). Solid lines represent the predictive mean. The confidence interval obtained with the Bayesian approach is not only grossly overconfident (contains less than $80\%$ of the total data) but also partially extremely loose, exhibiting unnecessarily large confidence intervals far away from the data. By contrast, our approach is both accurate and tight.

## 5    Proposed Approach

In this section, we present our approach to obtaining calibrated GPs by discarding the posterior variance and obtaining alternative predictive quantiles using a quantity inspired by the posterior variance and new hyperparmeters. As we will demonstrate, our approach has numerous advantages. First, by varying the hyperparameters used to obtain predictive quantiles, the resulting quantiles are sharper than what can be obtained simply by multiplying the posterior variance with an appropriate constant. Secondly, by exploiting the monotonicity of the hyperparameters, our approach can be used to obtain intervals for multiple confidence levels $\delta$ in a very efficient manner. Finally, our method is backed by tight theoretical guarantees, obtained by exploiting its connection to conformal prediction.

### 5.1    Sharp Calibrated GP for Single Confidence Level $\delta$

In the following, we describe how to obtain a sharply calibrated GP for a fixed desired calibration level $\delta$. Instead of scaling the GP posterior variance to meet the desired calibration level $\delta$, we propose computing predictive quantiles by training a new quantity similar to the posterior variance but with new hyperparameters. In other words, we discard the posterior variance of the first GP and replace it by a quantity that corresponds to the posterior variance of a different GP, which we train separately. This way, we allow for more degrees of freedom during calibration. We then leverage this additional freedom to minimize the distance of the quantiles to the predictive mean, which yields a sharply calibrated model.

We assume to have a data set $\mathcal{D}$, which we split into training data $\mathcal{D}_{\mathrm{tr}}$ and calibration (holdout) data $\mathcal{D}_{\mathrm{cal}}$, with $\mathcal{D} = \mathcal{D}_{\mathrm{tr}} \cup \mathcal{D}_{\mathrm{cal}}$. The training data $\mathcal{D}_{\mathrm{tr}}$ will be used to compute the posterior, whereas $\mathcal{D}_{\mathrm{cal}}$ will be used to calibrate the model. Note that while not splitting the data might be reasonable when using other types of regression models, e.g., Bayesian or ensemble neural networks, splitting the data in the case of GPs is beneficial for providing accurate quantiles for data out of distribution. This is because, for many commonly used kernels, the GP posterior distribution is considerably more concentrated for test points close to the training data $\mathcal{D}_{\mathrm{tr}}$ than for those far away from $\mathcal{D}_{\mathrm{tr}}$. Since we wish to obtain a model that is calibrated for data both close and far away from the training data, we take this into account during training by splitting the data. We also assume to have a *predictive* mean $\mu_{\mathcal{D}_{\mathrm{tr}}}(\boldsymbol{\theta}^R, \cdot)$, corresponding to GP posterior mean function, where the regressor hyperparameters $\boldsymbol{\theta}^R$ were obtained, e.g., via log-likelihood maximization, as discussed in Section 4. However, note that any other way of choosing the posterior mean hyperparameters $\boldsymbol{\theta}^R$ is permitted.

Given $\mu_{\mathcal{D}_{\mathrm{tr}}}(\boldsymbol{\theta}^R, \cdot)$ and $\mathcal{D} = \mathcal{D}_{\mathrm{tr}} \cup \mathcal{D}_{\mathrm{cal}}$, we then follow the convention of other recalibration approaches (Kuleshov et al., 2018; Marx et al., 2022) and aim to obtain, for an arbitrary $0 \le \delta \le 1$, a scalar $\beta_\delta$ and vector of hyperparameters $\boldsymbol{\theta}_\delta$, such that the corresponding predictive quantile $\mu_{\mathcal{D}_{\mathrm{tr}}}(\boldsymbol{\theta}^R, \cdot) +$

$\beta_\delta \sigma_{\mathcal{D}_{tr}}(\boldsymbol{\theta}_\delta, \cdot)$ contains $\delta$ times the total amount of data points. For this reason, we henceforth refer to $\boldsymbol{\theta}_\delta$ as *calibration hyperparameters*.

In order to obtain a sharply calibrated GP model, ideally we would like to choose $\beta_\delta$ and $\boldsymbol{\theta}_\delta$ such that they minimize the expected length of the centered intervals (2) subject to calibration. However, this optimization problem is hard to solve. Hence, we instead attempt to improve model sharpness by concentrating the predictive distribution around the predictive mean $\mu_{\mathcal{D}_{tr}}(\boldsymbol{\theta}^R, \cdot)$, i.e., by minimizing the deviation of the quantiles from $\mu_{\mathcal{D}_{tr}}(\boldsymbol{\theta}^R, \cdot)$. This corresponds to solving the optimization problem

$$
\begin{aligned}
&\min_{\substack{\beta_\delta \in \mathbb{R} \\ \boldsymbol{\theta}_\delta \in \boldsymbol{\Theta}}} \sum_{i=1}^{N_{cal}} \beta_\delta^2 \sigma_{\mathcal{D}_{tr}}^2\left(\boldsymbol{\theta}_\delta, \boldsymbol{x}_{cal}^i\right) \\
&\text{s.t.} \sum_{i=1}^{N_{cal}} \frac{\mathbb{I}_{\geq 0}\left(\Delta y_{cal}^i - \beta_\delta \sigma_{\mathcal{D}_{tr}}\left(\boldsymbol{\theta}_\delta, \boldsymbol{x}_{cal}^i\right)\right)}{N_{cal} + 1} = \delta
\end{aligned}
\tag{5}
$$

where $\left\{\boldsymbol{x}_{cal}^i, y_{cal}^i\right\} \in \mathcal{D}_{cal}$ are samples from the calibration data set, $\Delta y_{cal}^i := y_{cal}^i - \mu_{\mathcal{D}_{tr}}(\boldsymbol{\theta}^R, \boldsymbol{x}_{cal}^i)$ corresponds to the difference between predicted and measured output, and $N_{cal} = |\mathcal{D}_{cal}|$ to the number of data points used for calibration.

*Remark* 5.1. Note that we employ $N_{cal} + 1$ in the denominator in (5) instead of $N_{cal}$. Though this choice makes little difference in practice, we require it for theoretical guarantees.

The equality constraint in (5) is generally infeasible, e.g., if $N_{cal} = 2$ and $\delta = 0.5$, and is discontinuous, making it hard to solve. As it turns out, this can be easily remedied without any detriment to sharpness or calibration, and the problem can be rendered considerably easier to solve by substituting the equality constraint in (5) with

$$
\beta_\delta = q_{lin}(\delta, \boldsymbol{\Sigma}_{\mathcal{D}_{tr}}^{-1}\boldsymbol{\Delta y}_{cal}),
\tag{6}
$$

where $q_{lin}(\delta, \boldsymbol{\Sigma}_{\mathcal{D}_{tr}}^{-1}\boldsymbol{\Delta y}_{cal})$ is a monotonically increasing piecewise linear function[1] that maps $\delta = j/(N_{cal} + 1)$ to the $j$-th smallest entry of $\boldsymbol{\Sigma}_{\mathcal{D}_{tr}}^{-1}\boldsymbol{\Delta y}_{cal}$, and the entries of the vector

$$
\boldsymbol{\Sigma}_{\mathcal{D}_{tr}}^{-1}\boldsymbol{\Delta y}_{cal} = \left(\frac{\Delta y_{cal}^1}{\sigma_{\mathcal{D}_{tr}}\left(\boldsymbol{\theta}_\delta, \boldsymbol{x}_{cal}^1\right)}, ..., \frac{\Delta y_{cal}^{N_{cal}}}{\sigma_{\mathcal{D}_{tr}}\left(\boldsymbol{\theta}_\delta, \boldsymbol{x}_{cal}^{N_{cal}}\right)}\right)^\top
$$

correspond to the z-scores of the data under the calibration standard deviation $\sigma_{\mathcal{D}_{tr}}(\boldsymbol{\theta}_\delta, \cdot)$. The original problem (5) then becomes

$$
\min_{\boldsymbol{\theta} \in \boldsymbol{\Theta}} \sum_{i=1}^{N_{cal}} \left[q_{lin}(\delta, \boldsymbol{\Sigma}_{\mathcal{D}_{tr}}^{-1}\boldsymbol{\Delta y}_{cal})\sigma_{\mathcal{D}_{tr}}\left(\boldsymbol{\theta}_\delta, \boldsymbol{x}_{cal}^i\right)\right]^2,
\tag{7}
$$

which is considerably easier to solve due to the lack of constraints, and enables us to use gradient-based approaches, since $q_{lin}(\delta, \boldsymbol{\Sigma}_{\mathcal{D}_{tr}}^{-1}\boldsymbol{\Delta y}_{cal})$ is differentiable with respect to $\sigma_{\mathcal{D}_{tr}}(\boldsymbol{\theta}_\delta, \boldsymbol{x}_{cal}^i)$.

*Remark* 5.2. The choice of interpolant (6) is due to its simplicity. However, other forms of monotone interpolation are also possible. For high data sizes, the choice of interpolant becomes of little relevance, since we only perform small interpolation steps.

## 5.2 Calibrated GP for Arbitrary Confidence Level $\delta$

While (7) is useful for obtaining a sharply calibrated model for a single confidence level $\delta$, solving (7) multiple times whenever we want to obtain sharply calibrated models for different confidence levels $\delta$ can be time-consuming. Furthermore, interpolating between any two arbitrary solutions of (7) won't necessarily yield a result close to the desired calibration. Fortunately, we can leverage Assumption 4.1 to show that interpolating between two solutions of (7) will yield a result close to the

---

[1]For $\boldsymbol{a} \in \mathbb{R}^{N_{cal}}$, a permutation $i_1, \ldots, i_{N_{cal}} \in [1, ..., N_{cal}]$ where $a_{i_j} \leq a_{i_{j+1}}$ for all $j$, and $\delta \in [l/(N_{cal} + 1), (l+1)/(N_{cal} + 1)]$,

$$
q_{lin}(\delta, \boldsymbol{a}) = a_{i_l} + (\delta(N+1) - l)\left(a_{i_{l+1}} - a_{i_l}\right).
$$

---

**Algorithm 1** Training Calibration Hyperparameters for Arbitrary Confidence Level

---

**Input:** kernel $k(\cdot, \cdot, \cdot)$, predictor $\mu_{\mathcal{D}_{\mathrm{tr}}}(\boldsymbol{\theta}^R, \cdot)$, calibration data $\mathcal{D}_{\mathrm{cal}}$, confidence levels $\delta_1, ... \delta_N$
**for** $i = 1$ **to** $M$ **do**
    Compute $\beta_{\delta_1}, \boldsymbol{\theta}_{\delta_1}, ..., \beta_{\delta_{N_{\mathrm{cal}}}}, \boldsymbol{\theta}_{\delta_{N_{\mathrm{cal}}}}$ by solving (6) and (7) subject to

$$\boldsymbol{\theta}_{\delta_i} \leq \boldsymbol{\theta}_{\delta_j}, \quad \text{if } \delta_i < \delta_j \text{ and } \beta_{\delta_i} \geq 0,$$

$$\boldsymbol{\theta}_{\delta_i} \geq \boldsymbol{\theta}_{\delta_j}, \quad \text{if } \delta_i < \delta_j \text{ and } \beta_{\delta_j} \leq 0.$$

**end for**
Fit a continuous, monotonically increasing interpolation model $\hat{\beta}(\delta)$ and a continuous model $\hat{\boldsymbol{\theta}}(\delta)$ using the training data $\{\delta_i, \beta_{\delta_i}, \boldsymbol{\theta}_{\delta_i}\}_{i=1,...,N}$
**Output:** $\hat{\beta}(\delta), \hat{\boldsymbol{\theta}}(\delta)$

---

desired calibration, provided that we interpolate between two strictly increasing or decreasing sets of hyperparameters. Formally, this is achieved by solving (7) $N_{\mathrm{cal}} + 1$ times to obtain $\beta_0, \beta_{\delta_1}, ..., \beta_{\delta_N} \in \mathbb{R}$ and $\boldsymbol{\theta}_0, \boldsymbol{\theta}_{\delta_1}, ..., \boldsymbol{\theta}_{\delta_N} \in \boldsymbol{\Theta}$, subject to two additional constraints. First, the calibration scaling parameters $\beta_\delta$ must be monotonically increasing with $\delta$, i.e.,

$$\beta_{\delta_i} \leq \beta_{\delta_j} \quad \text{if } \delta_i < \delta_j,$$

and the calibration hyperparameters $\boldsymbol{\theta}_\delta$ must be decreasing with $\delta$ if $\beta_\delta$ is negative, and increasing if $\beta_\delta$ is positive, i.e.,

$$\boldsymbol{\theta}_{\delta_i} \leq \boldsymbol{\theta}_{\delta_j}, \quad \text{if } \delta_i < \delta_j \text{ and } \beta_{\delta_i} \geq 0,$$
$$\boldsymbol{\theta}_{\delta_i} \geq \boldsymbol{\theta}_{\delta_j}, \quad \text{if } \delta_i < \delta_j \text{ and } \beta_{\delta_j} \leq 0.$$

In other words, the entries of $\boldsymbol{\theta}_\delta$ are strictly decreasing with $\beta_\delta$ up until the sign of $\beta_\delta$ switches, after which they are increasing. The reason why we impose these restrictions is that we can then confidently interpolate between any values of $\beta_\delta$ and $\boldsymbol{\theta}_\delta$, since Assumption 4.1 implies that the quantile stipulated by $\beta_\delta \sigma_{\mathcal{D}_{\mathrm{tr}}}(\boldsymbol{\theta}, \cdot)$ is monotonically increasing with $\delta$. We then train simple piecewise linear interpolation models $\hat{\beta} : [0, 1] \to \mathbb{R}$ and $\hat{\boldsymbol{\theta}} : [0, 1] \to \boldsymbol{\Theta}$, such that $\hat{\beta}(\delta_i) = \beta_{\delta_i}$ and $\hat{\boldsymbol{\theta}}(\delta_i) = \boldsymbol{\theta}_{\delta_i}$, with the additional constraint that $\hat{\boldsymbol{\theta}}(\delta)$ reaches a minimum whenever $\hat{\beta}(\delta) = 0$, which can be potentially achieved by adding an artificial vector of training hyperparameters $\boldsymbol{\theta}_\delta$ for computing $\hat{\boldsymbol{\theta}}(\delta)$. Note, however, that any other form of monotone interpolation is also acceptable for obtaining $\hat{\beta}(\delta)$ and $\hat{\boldsymbol{\theta}}(\delta)$. The procedure is summarized in Algorithm 1.

*Remark* 5.3. While lengthscale constraints of the form $\boldsymbol{\theta}_{\delta_j} \leq \boldsymbol{\theta}_{\delta_{j+1}}$ can be easily enforced when solving (7) by substituting $\boldsymbol{\theta}$ with $\boldsymbol{\theta}_{\delta_j} + \Delta\boldsymbol{\theta}$ and minimizing over the logarithm of $\Delta\boldsymbol{\theta}$, the constraint $\beta_{\delta_j} \leq \beta_{\delta_{j+1}}$ is not enforced in (7). However, in practice we were often able to find local minima of (7) that satisfy this requirement.

We can then easily show that our approach achieves an arbitrary calibration level as the amount of data grows, provided that we choose the confidence levels $\delta$ accordingly.

**Theorem 5.4.** *Let $y$ be absolutely continuous conditioned on $\boldsymbol{x}$, let $\mu_{\mathcal{D}_{tr}}(\boldsymbol{\theta}^R, \cdot)$ be a posterior GP mean, and let $\sigma_{\mathcal{D}_{tr}}(\cdot, \cdot)$ be a GP posterior variance conditioned on $\mathcal{D}_{tr}$. Then, for any calibration data set $\mathcal{D}_{cal} = \{\boldsymbol{x}_{cal}^i, y_{cal}^i\}$, choose*

$$\delta_1 = \frac{1}{N_{cal} + 1}, \quad \delta_2 = \frac{2}{N_{cal} + 1}, \quad ..., \quad \delta_{N_{cal}} = \frac{N_{cal}}{N_{cal} + 1},$$

*and let $\hat{\beta}(\cdot)$ and $\hat{\boldsymbol{\theta}}(\cdot)$ be interpolation models obtained with Algorithm 1 and confidence levels $\delta_1, ..., \delta_{N_{cal}}$. Then*

$$\mathbb{P}_{\boldsymbol{x}, y \sim \Pi}\left(y - \mu_{\mathcal{D}_{tr}}(\hat{\boldsymbol{\theta}}(\delta), \boldsymbol{x}) \leq \hat{\beta}(\delta)\sigma_{\mathcal{D}_{tr}}(\hat{\boldsymbol{\theta}}(\delta), \boldsymbol{x})\right) \in \left[\delta - \frac{1}{N_{cal} + 1}, \delta + \frac{1}{N_{cal} + 1}\right].$$

*Proof.* The proof can be found in the supplementary material. □

Note that Theorem 5.4 also implies that a single set of calibration parameters $\beta_\delta$ and $\boldsymbol{\theta}_\delta$ obtained by solving (7), since we can substitute $\hat{\beta}(\delta) = \beta_\delta$ and $\hat{\boldsymbol{\theta}}(\delta) = \boldsymbol{\theta}_\delta$ into (8).

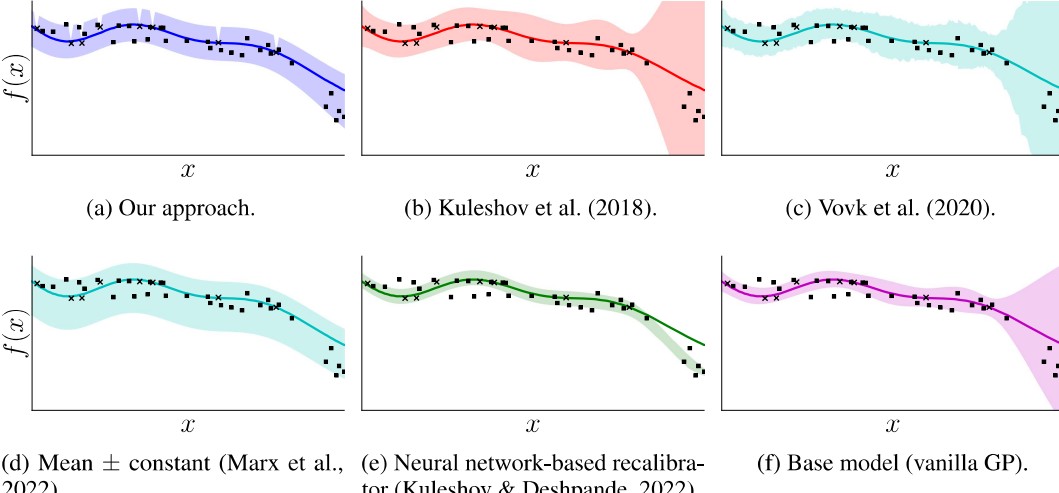

(a) Our approach.  (b) Kuleshov et al. (2018).  (c) Vovk et al. (2020).

(d) Mean $\pm$ constant (Marx et al., 2022).  (e) Neural network-based recalibrator (Kuleshov & Deshpande, 2022).  (f) Base model (vanilla GP).

Figure 2: Centered 99 % confidence intervals (shaded regions) obtained with our method, vanilla GPs recalibrated using the approaches of Kuleshov et al. (2018) and Vovk et al. (2020), a naive vanilla GP, the point-predictor (variance-free) approach proposed in Marx et al. (2022), and a naive fully Bayesian GP. Solid lines represent the predictive mean, crosses represent data used to train the base model, and squares represent calibration (holdout) data. Our approach yields a model that is both well-calibrated and sharp. This is because of the added flexibility that comes from being able to also change the lengthscale to design a calibrated model.

## 6 Discussion

**Computational Complexity.** Much like hyperparameter optimization for standard GPs (Rasmussen & Williams, 2006), the major driver behind the computational complexity in our approach stems from the need to invert the covariance matrix, an operation that scales cubically with the amount of data. In order to alleviate the computational cost of our approach, we can resort to different tools that improve scalability (Liu et al., 2020). One approach consists of employing only a subset of the training data $\mathcal{D}_{\text{tr}}$ to choose the calibration hyperparameters $\boldsymbol{\theta}_\delta$, and then the full data set to choose $\beta_\delta$. While this potentially leads to a loss in sharpness compared to when using the full data set, it still guarantees a calibrated model. Our technique is also readily applicable to sparse GPs (Snelson & Ghahramani, 2005; Titsias, 2009), as Assumption 4.1 typically still holds. This option is also explored in numerical experiments, in Section 7. Moreover, in many settings a specific level of calibration is often required, as opposed to several different ones, e.g., in stochastic model predictive control, where chance constraints corresponding to a fixed risk have to be satisfied (Mesbah, 2016). In such settings, we potentially only have to train a single vector of calibration hyperparameters $\boldsymbol{\theta}_\delta$, which reduces computational cost.

**Initialization and Solution.** The choice of initial hyperparameters can affect the optimization results considerably, and choosing a good hyperparameter initialization can be challenging, as is true when choosing the hyperparameters for the predictive mean. While this can be partially addressed by employing random restarts, we can also reuse trained models for similar calibration levels, since it is reasonable to expect that only small changes to the calibration hyperparameters are required to achieve a slight increase or decrease in confidence level. Furthermore, we can also simplify the problem by considering only a scaled version of the regression hyperparameters $\boldsymbol{\theta}^R$ to compute $\boldsymbol{\theta}_\delta$, which would reduce the optimization problem to a line search.

## 7 Experiments

In this section, we apply and analyze our approach using a toy data set and different regression benchmark data sets from the UCI repository. In the supplementary material, we also compare our

approach to that of Capone et al. (2022) when used to obtain uniform error bounds and apply our method to two different Bayesian optimization problems.

The goal is for our approach to obtain a sharp calibrated regression model for each data set in the calibrated regression experiments. We test our approach on various data sets and compare it to the state-of-the-art recalibration approaches by Kuleshov et al. (2018) and Vovk et al. (2020), the point predictor (posterior variance-free) approach proposed in Marx et al. (2022), as well as the check-score-based approach of Kuleshov & Deshpande (2022). The technique proposed by Kuleshov et al. (2018) essentially multiplies the vanilla posterior standard deviation $\sigma_{\mathcal{D}_{\text{tr}}}(\boldsymbol{\theta}^R, \cdot)$ with the recalibrated z-score, such that the confidence level observed on the calibration data matches that of the desired confidence level. Vovk et al. (2020) employ a similar approach, except that random interpolation is employed to compute new scaling values. The point predictor-based method proposed in Marx et al. (2022) discards the posterior standard deviation $\sigma_{\mathcal{D}_{\text{tr}}}(\boldsymbol{\theta}^R, \cdot)$ and computes a constant scalar that is added to the predictive mean and used to compute quantiles everywhere within the input space. The method of Kuleshov & Deshpande (2022) trains a neural network using a quantile loss, which takes base quantiles as inputs and returns new, recalibrated quantiles. As a recalibrator for the method of Kuleshov & Deshpande (2022), we employ the same neural network architecture suggested in their paper, trained over 200 epochs, with additional pretraining over 2000 epochs using a single dataset for the UCI experiments. In all experiments except kin8nm and Facebook comment volume 2, we employ standard GPs with automatic relevance determination squared-exponential (ARD-SE) kernels and zero prior mean as base models, trained using log-likelihood maximization. For the kin8nm and Facebook comment volume 2 datasets, we employ sparse GPs (Titsias, 2009) with zero prior mean, ARD-SE kernels, and 300 inducing points.

## 7.1 Toy Data Set

The first regression data set corresponds to a one-dimensional synthetic data set, where the results can be easily displayed visually. The main purpose of this section is to give an intuition as to how our approach computes confidence intervals compared to other techniques. We investigate the performance of our approach and compare it to other methods when employed to compute centered $99\%$ confidence intervals. We observe that the confidence intervals obtained with our approach peak less strongly far away from the data while being tight near the data compared to all other approaches except the one of Kuleshov & Deshpande (2022). This is because we allow the lengthscale to change to obtain a calibrated model. In contrast, all other methods except that of Kuleshov & Deshpande (2022) scale the standard GP posterior variance without changing hyperparameters. The method of Kuleshov & Deshpande (2022), which uses a neural network as a recalibrator, offers additional flexibility, resulting in sharper confidence intervals. However, calibration is not explicitly enforced during training, resulting in poor calibration. The results are depicted in Figure 2.

## 7.2 Benchmark Data Sets

We now experiment with seven different regression data sets from the UCI repository, two containing over eight thousand data points and requiring sparse GP approximations. The training/calibration/test split is 0.6, 0.2, and 0.2 for all data sets except the Facebook comment volume 2 data set, which contains over 80000 data points, and where the split is 0.08, 0.02, and 0.9. For the approach of Kuleshov & Deshpande (2022), we follow the steps in their paper and limit the calibration data size to 500.

We assess performance by employing diagnostic tools commonly used to assess calibration and sharpness (Kuleshov et al., 2018; Marx et al., 2022; Gneiting et al., 2007)). The score used to quantify calibration is the calibration error (Kuleshov et al., 2018), given by

$$\text{cal}\left(\mu_{\mathcal{D}_{\text{tr}}}(\boldsymbol{\theta}^R, \cdot), \beta_{\cdot}, \sigma_{\mathcal{D}_{\text{tr}}}(\cdot, \cdot)\right) = \sum_{j=1}^{m} \left(p_j - \hat{p}_j\right)^2, \tag{8}$$

where $p_j$ corresponds to the $j$-th desired confidence level, chosen, e.g., evenly spaced between 0 and 1, and $\hat{p}_j$ is the observed confidence level, i.e.,

$$\hat{p}_j = \frac{\left|\left\{y_t^* \mid \Delta y_t^* \leq \beta_{p_j} \sigma_{\mathcal{D}_{\text{tr}}}(\boldsymbol{\theta}_{p_j} \boldsymbol{x}_t^*), t = 1, ..., T\right\}\right|}{T}. \tag{9}$$

Table 1: Expected calibration error and sharpness of different methods over 100 repetitions per experiment. We report the expected calibration error (ECE), the average predictive standard deviation (STD), negative log-likelihood (NLL) and 95% confidence interval width (95% CI) obtained with our approach, vanilla GPs recalibrated using the methods of Kuleshov et al. (2018) (RK) and Vovk et al. (2020) (RV), the variance-free approach proposed in Marx et al. (2022) (RM), and the neural network-based recalibrator of Kuleshov & Deshpande (2022) (NN). We additionally report performance for the base model (B), which corresponds to a vanilla GP without the holdout data. Lower is better for all metrics. In all experiments except the Facebook2 dataset, our method is sharpest compared to all other methods except that of Kuleshov & Deshpande (2022). However, Kuleshov & Deshpande (2022) performs more poorly in terms of expected calibration error.

| DATA SET | METRIC | OURS | RK | RV | RM | NN | B |
|----------|--------|------|-----|-----|-----|-----|---|
| BOSTON | ECE | 0.003 | **0.0029** | **0.0029** | **0.0029** | 0.0056 | 0.041 |
| | STD | **0.16** | 0.31 | 0.3 | 0.33 | 0.22 | 1.9 |
| | NLL | 0.21 | 0.39 | 0.4 | 0.42 | **-0.24** | 1.6 |
| | 95% CI | 0.76 | 1.4 | 1.4 | 1.4 | **0.73** | 7.4 |
| YACHT | ECE | 0.0044 | **0.0043** | 0.0044 | **0.0043** | 0.0081 | 0.039 |
| | STD | 0.16 | 0.5 | 0.47 | 0.5 | **0.14** | 2.8 |
| | NLL | 0.26 | 0.68 | 0.69 | 0.68 | **-2** | 2 |
| | 95% CI | 0.76 | 2.3 | 2.3 | 2.3 | **0.3** | 11 |
| MPG | ECE | 0.0036 | **0.0035** | **0.0035** | **0.0035** | 0.0053 | 0.044 |
| | STD | **0.13** | 0.38 | 0.38 | 0.38 | 0.29 | 2.8 |
| | NLL | 0.032 | 0.63 | 0.64 | 0.63 | **0.02** | 2 |
| | 95% CI | **0.6** | 1.7 | 1.8 | 1.7 | 0.96 | 11 |
| WINE | ECE | **0.00047** | **0.00047** | **0.00047** | **0.00047** | 0.0067 | 0.0058 |
| | STD | **0.54** | 1 | 1 | 0.88 | 0.72 | 1.4 |
| | NLL | 1.2 | 1.3 | 1.3 | 1.3 | **-0.36** | 1.4 |
| | 95% CI | **2.1** | 3.8 | 3.8 | 3.9 | 2.8 | 5.4 |
| CONCRETE | ECE | 0.00071 | **0.00064** | **0.00064** | **0.00064** | 0.00076 | 0.032 |
| | STD | **0.25** | 0.64 | 0.64 | 0.64 | 0.57 | 2.8 |
| | NLL | **0.72** | 1.1 | 1.1 | 1.1 | 0.85 | 2 |
| | 95% CI | **0.93** | 2.5 | 2.5 | 2.5 | 2.1 | 11 |
| KIN8NM | ECE | **0.00016** | **0.00016** | **0.00016** | **0.00016** | 0.00053 | 0.028 |
| | STD | **0.074** | 0.12 | 0.12 | 0.12 | 0.098 | 0.4 |
| | NLL | -0.54 | -0.65 | -0.65 | -0.63 | **-0.76** | 0.1 |
| | 95% CI | **0.26** | 0.47 | 0.47 | 0.48 | 0.44 | 1.6 |
| FACEBOOK2 | ECE | 0.00044 | **0.00043** | **0.00043** | 0.00045 | 0.0089 | 0.044 |
| | STD | **0.068** | 0.18 | 0.18 | 0.18 | 0.18 | 1.2 |
| | NLL | 3.6 | -1.3 | -1.3 | -1.2 | **-2.3** | 1.2 |
| | 95% CI | **0.6** | 1.7 | 1.7 | 1.7 | 3.4 | 4.6 |

Here the superscript $*$ denotes test inputs and outputs, $T$ denotes the total number of test points, and $\Delta y_t^* := \mu_{\mathcal{D}_{tr}}(\theta^R, x_t^*) - y_t^*$. We employ $m = 21$ evenly spaced values between 0 and 1 for $p_j$. To measure sharpness, we employ the average length of the 95% confidence interval, the average standard deviation of the predictive distribution, and the average negative log-likelihood of the predictions (Gneiting et al., 2007; Marx et al., 2022). Note that since every model outputs a quantile for any desired calibration level, the corresponding negative log-likelihood and average standard deviation are well specified. These are computed by employing the cumulative distribution function, obtained by inverting the quantile function specified by each model.

We carried out each experiment 100 times and report the resulting average expected calibration error, standard deviation, negative log-likelihood, and length of the centered 95% confidence intervals in Table 1. Our approach performs best or marginally worse than all other calibration approaches regarding expected calibration error. This is to be expected from Theorem 5.4. Furthermore, it outperforms all approaches except that of Kuleshov & Deshpande (2022) in sharpness. However, the improved sharpness of the method of Kuleshov & Deshpande (2022) comes at the expense of calibration.

# 8 Conclusion

We have presented a calibration method for Gaussian process regression that leverages the monotonicity properties of the kernel hyperparameters to obtain sharp calibrated models. We show that, under reasonable assumptions, our method yields an accurately calibrated model as the size of data used for calibration increases. When applied to different regression benchmark data sets, our approach was shown to be competitive in sharpness compared to state-of-the-art recalibration methods without sacrificing calibration performance. It is worth stressing that, though the tools presented here emerge naturally from a Gaussian process setting, we do not require our predictor to be a Gaussian process to obtain theoretical guarantees. In future work, we aim to leverage similar monotonicity characteristics to get sharply calibrated models using tools different from Gaussian processes. Furthermore, we aim to experiment with inducing variables as hyperparameters when optimizing the models for sharpness.

## Acknowledgements

This work was supported in part by the European Research Council Consolidator Grant Safe data-driven control for human-centric systems (CO-MAN) under grant agreement number 864686.

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
