# OpenReview forum: "Sharp Calibrated Gaussian Processes"
_NeurIPS.cc/2023/Conference — NeurIPS 2023 poster_

### Official Review · Reviewer_k7Z4 · 2023-07-03

**Soundness:** 2 fair
**Presentation:** 3 good
**Contribution:** 3 good
**Rating:** 5
**Confidence:** 3

**Summary:**

Motivated by the observation that the posterior variance of a Gaussian process is often poorly calibrated, the authors propose an alternative approach of attaching predictive quantiles to the posterior mean. In essence, their approach minimises the width of the predictive quantiles under an empirical calibration constraint computed on held-out validation data. By satisfying the empirical calibration constraint, the authors prove that the predictive quantiles are indeed approximately the right quantiles (Theorem 5.3). The approach is tested and compared against other calibration approaches in a toy example and on seven data sets from the UCI data set repository. The results show that it outperforms other techniques in terms of sharpness.

**Strengths:**

To begin with, I would like to thank the authors for their submission.

## Strengths

* The paper is easy to follow and generally well written. I found only a few typos.

* The problem that the posterior variance of a GP may be poorly calibrated is highly relevant, so methods that attempt to attack this problem, like the proposed approach by the authors, are certainly important.

* Accompanying the method with a theoretical result that guarantees correctness (Theorem 5.3) reassures the practitioner.

* I really like Section 5.2, where the authors spend some effort on reducing the computational cost in practice. They could have just left it at (7) and state that the optimisation problem would have to be computed for every confidence level $\delta$ of interest, but of course the approach in Section 5.2 is much more elegant.

* The experiments appear to show that the proposed approach is sharper than alternatives. I, however, have some doubts about the experimental section. Please see below.

## General Comments and Suggestions

* Should the title of the paper be "Sharply Calibrated GPs" instead of "Sharp Calibrated GPs"? I think "Sharp(ly)" should be an adverb, since it modifies "calibrated".

* On line 75, you use $\mathcal{D}\_{\text{tr}}$ without introducing the symbol first. This is confusing, because you have just introduced $\mathcal{D}$, not $\mathcal{D}\_{\text{tr}}$, as the training data.

* On line 75, you introduce the cut-point term as $\beta_\delta \sigma_{\mathcal{D}}(\delta, x)$. At this point, you should really better explain this term by answering the following questions: Should $\sigma_{\mathcal{D}}(\delta, x)$ be interpreted as a standard deviation? If so, how do you make sense of that it depends on $\delta$? If not, then what is $\sigma$? If you're learning $\sigma$, why do you also need $\beta_\delta$? Can you not absorb $\beta_\delta$ into $\sigma$? You should also mention that $\beta_\delta$ may be negative. Without that information, (1) doesn't make much sense for small $\delta$.

* On line 81: Do you mean "are small" or "are as small as possible"? This nuance is very important and changes the meaning substantially.

* On line 119, you say that the log-marginal likelihood does not account for calibration. Depending on what precisely you mean by calibration, I think that this is false: the log-marginal likelihood is an empirical estimate of the KL divergence, and the KL divergence certainly accounts for the "whole distribution" and therefore the calibration.

* In Section 5.1, is $\sigma\_{\mathcal{D}\_{\text{tr}}}(\theta, x)$ the posterior variance of the GP where the kernel parameters are now the parameters that we optimise over? From line 146 onwards, you don't actually explain this! Since this is a crucial part of your construction, I think the exposition would be better if you were to clearly explain this somewhere around line 158.

* On line 175, you state that (5) can be replaced by (6) with further explanation. Since this is an important step in the derivation of your algorithm, I think that it deserves a careful explanation. Moreover, it is not true that (5) and (6) are equivalent, since $q\_{\text{lin}}$ is linearly interpolated. I think the exposition would benefit from a little more care here.

* Line 268: "calibration" -> "calibrated"

* Could you add to the legend of Figure 2 what the squares and crosses are? Why does it look like there are two "lines of squares/crossed"?

**Weaknesses:**

## Weaknesses

### Assumption 4.1 Not Obviously Satisfied

I agree that Assumption 4.1 is obviously satisfied for the variance of a kernel. However, for the inverse length scale, I can believe that Assumption 4.1 might be satisfied, but this is not obvious at all. Would the authors be able to produce a proof that the posterior variance of a GP is monotonic in the inverse length scale?

### Theorem 5.3 Might Not Be Valid

The proof of Theorem 5.3 crucially relies on Theorem 1 by Marx et al. (2022). This Theorem 1, however, operates in the setting where all pairs $(x_i, y_i)$ are sampled i.i.d. (see Section 2 of Marx et al., 2022). But in the setting of GPs, which is the setting of the submission, the pairs aren't independent, because they are correlated by the sample from the underlying GP! This means that Theorem 1 might not apply, which means that Theorem 5.3 might not be valid. Could the authors comment on this?

### Result of Section 7.1 Looks Questionable

In Figure 2, you state that your approach produces a 99% confidence interval. However, if you look at the right side of Figure 2.(a), then the confidence interval completely misses the data! It hence looks like the shaded region is not actually a 99% confidence interval, which makes me wonder whether the predictions are actually well calibrated.

### Bolded Results Are Not Significantly Best Results

Throughout the main paper and the supplement, you bold the score with the best average. However, if $x_1 \pm e_1$ and and $x_2 \pm e_2$ are such that $x_1 < x_2$, but $x_1 + e_1 \ge x_2 - e_2$, then you cannot actually conclude with confidence that $x_1 \pm e_1$ is really lower than $x_2 \pm e_2$, because the difference might be explained by random fluctuations. In other words, you should really only bold results that are the best results at reasonable statistical significance. Currently, because of this issue, I think that the results throughout are misrepresented.

### What Are STD and NLL in Table 2.1?

The premise of the paper is that you discard the predictive variance and instead produce calibrated predictive quantiles at one or multiple given confidence levels. This means that the predictions now consist of a mean and associated intervals. Therefore, the predictions are no longer probabilistic, so I really don't understand what the STD and NLL in Table 2 are! (For given a mean and an interval, how can you compute a probability?)

**Questions:**

## Conclusion

Although I think the proposed approach is very interesting, I am mainly worried about the validity of Theorem 5.3, the soundness the result in Section 7.1, the use of boldness to present the experimental results, and the STD and NLL metrics in Table 2.1. Therefore, at this point, I cannot accept this submission and must unfortunately recommend rejection. However, if the authors are able to address the following points, then I am willing to change my reject to an accept:

* Please see if Theorem 5.3 is really flawed, and, if it is, whether it can be fixed.

* Please correct the use of boldness throughout the main paper and the supplement.

* Please investigate whether the predictions in Figure 2.(a) really are calibrated.

* Please explain what the STD and NLL metrics in Table 2.1 are.

EDIT

The authors have largely addressed my criticisms, so I have increased my score to a borderline accept. I believe that the submission might deserve a higher score, but I am not comfortable recommending any higher without seeing a revised version of the PDF.

**Limitations:**

See above.

---

> ### Author Rebuttal · Authors · 2023-08-02
>
> Thank you very much for your review and your helpful comments. Please find our answers to your questions below. If you feel that we have adequately addressed your concerns and questions, we would appreciate if you would consider updating your score.
>
> **Assumption 4.1.** Capone et al. (2022) have shown that Assumption 4.1 holds for the lengthscale of a class of stationary kernels (Lemma 3.3). In fact, the proof presented therein can be employed to show that Assumption 4.1 holds for any hyperparameters that lead to a monotonous increase in the Fourier transform. We will include these details in the revision, and give thorough examples of hyperparameters that satisfy these properties. Note also that several results that employ the so-called "fill distance" show that Assumption 4.1 holds for a broad class of kernels as the lengthscale becomes very large or very small (see, e.g., Chapter 11 in *Scattered Data Approximation* by Holger Wendland, 2011).
>
> **Theorem 5.3.** We believe there may be some confusion regarding our assumptions as well as the requirements of Theorem 5.3. In a standard Bayesian setting, a GP prior would, as you point out, result in joint dependence amongst the observations. However, we note that our method is inherently frequentist in nature: we use GPs simply as mechanisms to produce kernelized mean functions and confidence intervals, rather than as a true Bayesian prior. Because of this frequentist setting, we make the standard frequentist assumptions about data: there is some unobserved (but fixed) function $f$, and $y \mid f(\boldsymbol x)$ is an i.i.d. distribution, meeting the requirements of Theorem 1 from Marx et al. (2022). Again, we emphasize that our model's mean and confidence intervals---which follow the same functional form as a Bayesian GP posterior---are purely for computation and do not arise from any assumed prior distribution.
>
> We will make this more clear in the revision, and we will explicitly specify in Section 2 that we make the assumption of i.i.d. input/observation pairs.
>
> **Use of bold font.** We have rewritten the tables and changed the exposition of the results; see the included PDF. In particular, we now only use bold font whenever the results are best statistically. Furthermore, we employ mean plus minus standard error, which best illustrates statistical significance in this case, and state this clearly in the revised version.
>
> **Section 7.1/Figure 2.** The main purpose of the toy example was to illustrate differences amongst how each method generated confidence intervals, as opposed to showing how well-calibrated they are. To this end, we deliberately picked a setting with little data and training inputs that are far apart, as this is where the difference between approaches is most obvious. Note that, in this setting, our method has no calibration guarantees, as our guarantees only hold in regions sufficiently near training data (i.e. in regions where test data and train data are i.i.d. draws from the same data distribution). This has caused some confusion, so we have revised the toy example. The toy example is now calibrated and also highlights the differences between approaches. Please refer to the submitted PDF for details.
>
> **STD and NLL in Table 2.1.** Thank you for this comment. Following the method presented in Section 5.2, we can use our method to compute predictive quantiles for arbitrary $\delta$. Hence, our model implicitly specifies a cumulative distribution function, obtained by inverting the quantile function. This allows us to compute the standard deviation and the negative log likelihood of the predictions.

---

> > ### Comment · Reviewer_k7Z4 · 2023-08-10
> >
> > Thank you for your reply to my rebuttal. I appreciate the clarifications. I am largely happy with the adjustments and will increase my score, as promised.
> >
> > I would like to make two final remarks:
> >
> > 1. In the attached PDF, Figure 1(c) for the regular GP looks suspicious: Did you all the noise variance to the predictions? Did you maximise the marginal likelihood and condition the GP on all the data? The uncertainty regions currently suspiciously increase, which makes me wonder whether the GP is really conditioned on all the data in the plot.
> >
> > 2. I would appreciate it if you could add a clarification in the main body about how the STD and NLL are computed in the experiments.

---

> > > ### Author Response · Authors · 2023-08-10
> > >
> > > Thank you very much for your comments. We are very happy to see that we were able to adequately address your questions.
> > >
> > > Regarding Figure 1c: the figure shows training and calibration data and the GP is not conditioned on all the data, hence the changes in uncertainty. This is described in Section 5.1 of the paper: we separate training from calibration data, allowing the GP to be calibrated on data that is also out of distribution.

---

> > > > ### Comment · Reviewer_k7Z4 · 2023-08-11
> > > >
> > > > Ah, this is interesting. I wonder whether this is a fair procedure.
> > > >
> > > > For calibrated GP methods, it makes sense to separate out the calibration data. On the other hand, for the regular GP, which doesn't have a calibration procedure, IMO it is most fair to learn and condition on the union of the training and calibration data.
> > > >
> > > > Could you comment on this?
> > > >
> > > > I now have an important question: Did you use a similar procedure for the experiments in the paper? That is, do the calibrated methods use an additional calibrated set that the vanilla GP was not conditioned on? If so, I now wonder how the vanilla GP would perform in the experiments in the case where it would be conditioned on all the data.

---

> > > > > ### Author Response · Authors · 2023-08-11
> > > > >
> > > > > Thank you for the interesting question.
> > > > >
> > > > > The premise of the vanilla GP is that it fully believes the prior. In other words,  it assumes the GP prior is the correct distribution for any data observed in the future, and does not change its prior based on the observed data. If the prior distribution indeed were correct, then the GP should be perfectly calibrated from the start, and conditioning on the observed data should have no effect on calibration performance. However, we should see an increase in sharpness, since the variance decreases around the data.
> > > > >
> > > > > In the more general frequentist setting, considered in our paper, it is difficult to say whether conditioning is better/fairer than not conditioning - although there should be an increase in sharpness by conditioning the GP on additional data, it is generally impossible to say whether calibration will improve or deteriorate, since the prior is always incorrect. That said, we ran additional experiments for the Boston, Yacht and Auto MPG datasets, and observed that sharpness and calibration improved, albeit only slightly, for the vanilla GP. However, these changes were largely insignificant compared to the results reported in the paper, and the vanilla GP is still significantly worse than other approaches in terms of calibration. We will add the results and discussion in the revision.

---

> > > > > > ### Comment · Reviewer_k7Z4 · 2023-08-12
> > > > > >
> > > > > > Thank you for the reply.
> > > > > >
> > > > > > Regarding the figure, I think you need to be careful here, because you don't want to mislead the readers. What precisely is it that you want to show? I think the figure would benefit from visually depicting which data points are train data, calibration data, and held-out data. Moreover, could you double check whether you've added the observation noise to the predictions of the GP? The predictions in the attached PDF look like you haven't.
> > > > > >
> > > > > > > We will add the results and discussion in the revision.
> > > > > >
> > > > > > Thank you for adding in these additional results and discussion. That is very helpful.

---

> > > > > > > ### Author Response · Authors · 2023-08-13
> > > > > > >
> > > > > > > Thank you very much for this remark.
> > > > > > >
> > > > > > > When displaying the results obtained with the vanilla GP, we aim to show how the base model performs without additional recalibration. In order to stress this and avoid confusion, we will relabel the vanilla GP as "Base model (uncalibrated)" in the revision and clearly state how it is computed in the text and captions. We will also clearly state that the calibration data is not used to train the base model but only for calibration when using our or any other recalibration approaches.
> > > > > > >
> > > > > > > Regarding the observation noise, we employed the base GP model from the GPyTorch toolbox, which does not add the variance of the observation noise to that of the posterior GP when computing predictive quantiles. The observation noise is only employed to compute the regularized covariance matrix, which is inverted when computing the GP posterior mean and standard deviation. However, we note that our approach and all other approaches considered in the experimental section can accommodate either setting, and this change is generally somewhat minor, since the noise standard deviation is often small compared to the signal standard deviation. To make sure, we reran several experiments where we added the noise variance to the posterior variance and can confidently state that including this change does not lead to a qualitative change in the results, mostly because of how small the noise parameter is. Furthermore, we observed that Figure 1c changes very little if this change is included due to how small the noise standard deviation is (the changes are hardly perceptible at first and can only be observed by zooming in).

---

> > > > > > > > ### Comment · Reviewer_k7Z4 · 2023-08-14
> > > > > > > >
> > > > > > > > > When displaying the results obtained with the vanilla GP, we aim to show how the base model performs without additional recalibration.
> > > > > > > >
> > > > > > > > That makes sense. The reason why I'm suggesting to visually distinguish the calibration data is that the reader might infer that the vanilla GP's uncertainty inflates around the these data, whereas the other methods do not. This is true, but in my opinion not an entirely fair comparison, because the calibrated procedures have "seen" the calibration data (through the calibration step).
> > > > > > > >
> > > > > > > > Importantly, you want to guard the reader from concluding that the uncertainy of vanilla GPs might not appropriately contract around observed data, as the figure in the attached PDF seems to suggest.
> > > > > > > >
> > > > > > > > >  However, we note that our approach and all other approaches considered in the experimental section can accommodate either setting, and this change is generally somewhat minor, since the noise standard deviation is often small compared to the signal standard deviation.
> > > > > > > >
> > > > > > > > Thank you for double checking this. We note that the observation noise is part of the prior and must always be added to the predictions to explain all variance in the observations. I'm glad to hear that it didn't make a big difference in these experiments, since the learned noise variances were small. However, in general it is not true that the learned noise variances are small, and in those cases leaving out the noise variance will make a drastic difference.
> > > > > > > >
> > > > > > > > > Furthermore, we observed that Figure 1c changes very little if this change is included due to how small the noise standard deviation is (the changes are hardly perceptible at first and can only be observed by zooming in).
> > > > > > > >
> > > > > > > > The subtle telltale of leaving off the observation noise is that the uncertainty regions produced by the predictions don't capture "most data". The uncertainty regions certainly definitely shouldn't strictly capture all observed data, as they are only $3\sigma$-regions, but a good rule of thumb is that they should capture most. This is what prompted me to asked whether you did.

---

> > > > > > > > > ### Author Response · Authors · 2023-08-15
> > > > > > > > >
> > > > > > > > > Thank you very much for your response.
> > > > > > > > >
> > > > > > > > > Since the vanilla and fully Bayesian GPs perform slightly better with the full dataset, we will use the full data set to train them when comparing against our method. This will be clearly stated and discussed in the revision. Regarding Figure 1c, this figure will be presented exclusively as the base model without calibration or additional training. We will then use a new, separate figure for the vanilla GP with the full data set in the toy example. We will also clarify the differences for the reader by pointing out that one model uses the full data set for training, whereas the calibration data set has been left out for the other.
> > > > > > > > >
> > > > > > > > > We agree that there are cases where the noise standard deviation can be very large, leading to potentially strong differences in results when employing it or leaving it out for predictions. We also agree that using the noise standard deviation is the correct way of predicting observations with the (purely Bayesian) vanilla GP. We originally treated the vanilla GP as if it had been trained on noisy data, but was predicting a function sampled from the prior GP (without noise). Please note that our approach and all other recalibration approaches, which are the focus of the paper, accommodate either setting due to their frequentist nature, so this choice is acceptable for recalibration. However, we realize that this can be confusing since we also compare our approach to the purely Bayesian setting, where the noise standard deviation should be added for predicting observations. Hence, in the revised paper, we will also use the noise standard deviation when performing predictions instead of using only the GP posterior standard deviation. We again stress that, in all our experiments, the noise standard deviation was always small after training the vanilla GP, leading only to small differences in results.

---

> > > > > > > > > > ### Comment · Reviewer_k7Z4 · 2023-08-15
> > > > > > > > > >
> > > > > > > > > > Thanks again for your comment. Everything you say sounds very reasonable, and the suggested amendments will improve the clarity of the exposition. I don’t have any  remaining questions, so I would like to thank you for engaging with me during the discussion period.

---

> > > > > > > > > > > ### Author Response · Authors · 2023-08-17
> > > > > > > > > > >
> > > > > > > > > > > Thank you kindly for engaging with us during the discussion phase.

---

### Official Review · Reviewer_XbYQ · 2023-07-05

**Soundness:** 2 fair
**Presentation:** 1 poor
**Contribution:** 3 good
**Rating:** 6
**Confidence:** 4

**Summary:**

The authors propose a novel method for calibrating Gaussian process posterior variances using held-out data. In particular, they train a new, separate, GP using the held out data for the variance, using the GP trained on the original dataset for the mean. They do this in a way which approximately maximises the sharpness (roughly speaking, minimises the variance). The calibration properties of the method are supported by theory. This is supported by an empirical evaluation of the calibration error on synthetic data and of the sharpness on real-world data.

Post-discussion: The authors have provided considerable improvement in their presentation of the results, and have added some extra results which further improve the clarity.

**Strengths:**

1. (major) the idea is novel,  and appears to be theoretically well supported, although I did not review the proofs in the supplementary material.
2. (major) The main claimed advantage of the proposed method is that is produces sharper predictives than comparable methods, which is well supported by the real-world experimental results, not withstanding the points raise in questions.
3. (major) Code is provided, which should improve reproducibility. It is based on a widely used framework, which increases potential for impact.

**Weaknesses:**

1. (major) The paper is not sufficiently clear. The method is fairly well explained, but the evaluation is extremely hard to follow, details of which are in the questions section. One particular area for straightforward correction is with Table 2: the values with the lowest mean are highlighted but there are several rows where the estimates are overlapping, so this should be clarified by highlighting.

**Questions:**

1. It's not clear to me how to interpret figure 2. I guess that the crosses are training data, what about the squares? Are they the calibration data?
2. I guess Table 1 is for the synthetic data plotted in figure 2, but I don't think this is stated anywhere.
2. When you train the vanilla GP, do you add the calibration data to the training data?
3. The sharpness of the predictives is claimed to be an advantage, but really it looks like you cover less of the data as a result in the toy example?
4. I appreciate that the main purpose of Table 2 is to compare the sharpness amongst the well-calibrated methods,  but I think it would be highly worthwhile to see how a vanilla GP compares on these same metrics.
5. With respect to line 103, it is mentioned that the assumption holds for isotropic kernels. I think that it also works out if you transform an isotropic kernel to have different lengthscales in each dimension, and I think you've implied this elsewhere, is that right?

Some suggestions and typos:
* For the hyperparameters, could you use $θ$?
* It would be helpful to have a bit more explanation in section 3. For example, instead of saying 'are small for every $δ$' perhaps you could say that the goal is to minimise that quantity for every $δ$.
* line 127: the z-score -> z-score
* Summations in equations 5, 7: provide the starting value (i=1 I guess)

**Limitations:**

The main limitation of the method appears to be its applicability to different covariance functions, and the added computational cost, both of which are noted clearly.

---

> ### Author Rebuttal · Authors · 2023-08-02
>
> Thank you for your review. We have made several improvements to the paper, particularly in the experimental section, to address your concerns about clarity. Below you can find our answers to your questions and comments. Corresponding modifications can also be seen in the PDF submitted with this rebuttal. Should you feel that the steps undertaken to clarify the paper address your concerns, we would be very grateful if you would consider raising your final score.
>
> **Presentation of numerical results.** We have removed bolding in Tables 1 and 2, except for cases that are statistically best-performing. The new tables show the mean plus minus standard error and we clearly indicate this in the revised version.
>
> **Responses to questions.**
>
> 1. We have modified Figure 2 and the toy example. The main purpose of the toy example was to illustrate differences amongst how each method generated confidence intervals, as opposed to showing how well-calibrated they are. To this end, we deliberately picked a setting with little data and training inputs that are far apart, as this is where the difference between approaches is most obvious. However, we do not expect our method to be calibrated in this low-data region, as our method is only guaranteed to work in regions near training data (i.e. we assume that the test data and train data are drawn i.i.d. from the same distribution).
>
> We realize that this setting did not clearly illustrate the calibration properties of each method, which made Figure 1 confusing. In the submitted PDF, we include a revised Figure 1 that uses a different toy dataset to better illustrated the calibration differences between each approach. Please refer to the submitted PDF for details.
>
> 2. Table 1 only summarizes the average calibration error for all methods and datasets. A full table with detailed experimental results can be found in the supplementary material (it did not fit in the main text). We now state this explicitly in the table caption. Furthermore, we have moved Figure 2 to a different page from that of Table 1 to underscore that Table 1 is unrelated to Figure 2.
>
> 3. The training data is not added to the vanilla GP. This is now stated explicitly in the experimental section.
>
> 4. Please refer to point 1.
>
> 5. The GP posterior standard deviation is considerably small in some examples, which leads to low STD values. However, this of course comes at the cost of a very poor calibration score, as can be seen in Table 1. We felt that including this data in the tables in the main paper would likely cause some confusion, since the models in this case are simply overconfident, not sharp AND calibrated. We will include the scores for the vanilla GP and fully Bayesian GP in the supplementary material. We have also included this information in the PDF submitted with the rebuttal.
>
> 6. Thank you for this observation. The statement indeed holds for the more general class of stationary kernels, i.e., kernels where k(x,y) = K(x-y) for some function K. We have rewritten the statement accordingly in the revised version.

---

> > ### Comment · Reviewer_XbYQ · 2023-08-14
> >
> > Apologies for the slow response.
> >
> > I am mostly satisfied, and I think the new figure is much clearer. There are just two points I want to follow up on.
> >
> > 3 - Did you mean to write "calibration data" instead of "training data" here? I think that you should add the calibration data to the training data when training the vanilla GP (or with the fully Bayesian approach), because you (I guess) lose something by keeping some data back for calibration. I guess that this would not make much difference to your results.
> >
> > 5 - I do not think the results made it into the attached pdf ... ? You could add a markdown table in a comment.
> >
> > I think that this is generally very good work, and I am keen to raise the score, but I just need these points clarified.

---

> > > ### Comment · Area_Chair_ekLD · 2023-08-14
> > > **Please close the loop on this information**
> > >
> > > Dear authors and reviewers, thanks for engaging. It'd be great if the authors could answer the reviewer ASAP, so there is time to reconsider the score.

---

> > > ### Author Response · Authors · 2023-08-15
> > >
> > > Thank you very much for your response and additional comments. We are very happy to see that we could address your previous comments adequately.
> > >
> > > 3 - Thank you for this observation. The squares denote both training and calibration data. We initially decided not to differentiate between both because we wanted to emphasize calibration quality over training quality. However, we realize this might be somewhat unclear, so we will use different symbols for the training and calibration data and clearly state this in the revision. Regarding the comparison with the vanilla and fully Bayesian GPs, it is safe to say that adding more data will improve sharpness since model confidence increases. However, it is generally difficult to say if calibration will improve, as the models always fully trust the (generally incorrect) GP prior. To test this, we reran several experiments in the setting where all data is used both for the vanilla and fully Bayesian GP, and observed that both calibration and sharpness improved, albeit only slightly. For this reason, in the revision, we will present the results for the vanilla and fully Bayesian models where all the available data is used to train them. This will be clearly stated and discussed. Furthermore, Figure 1c will be used exclusively to show how the base model performs before recalibration occurs and will be relabeled as "Base model (uncalibrated)".
> > >
> > > 5 - We accidentally omitted the results when generating the PDF. We apologize for this oversight. Below you can find the sharpness results (mean ± standard error) for the vanilla and Fully Bayesian GP, alongside ours. Please note that we do not have the data for the STD metric of the vanilla GP in the Facebook2 setting anymore and have not been able to rerun the dataset due to time constraints. This will be included in the revision.
> > >
> > >
> > > **Negative log-likelihood:**
> > > | | Boston  |  Yacht  | Auto MPG | Wine  | Concrete  | Kin8nm | Facebook2 |
> > > |---|---|---|---|---|---|---|---|
> > > | Ours | 0.24 ± 0.1 |  0.6 ± 0.02  | 0.5 ± 0.01| 1.5 ± 0.07 | 0.91 ± 0.04 |-0.56 ± 0.0 | -1.2 ± 0.03 |
> > > | Vanilla GP | 0.73± 0.04  |  0.85± 0.06   | 0.5± 0.04 | 1.5± 0.03 | 1.2± 0.02 |-0.51± 0.02 | 0.51± 0.02 |
> > > | Full Bayes |                       |  -1.2± 0.1  | 0.64± 0.2 |    |   |   |
> > >
> > > **Centered 95% intervals:**
> > > | | Boston  |  Yacht  | Auto MPG | Wine  | Concrete  | Kin8nm | Facebook2 |
> > > |---|---|---|---|---|---|---|---|
> > > | Ours | 1.2 ± 0.01 |  1.8 ± 0.03  | 1.6 ± 0.02| 4.7 ± 0.04 | 2.5 ± 0.04 |0.5 ± 0.02  | 1.5 ± 0.02 |
> > > | Vanilla GP | 3.2± 0.01  |  1.8± 0.05   | 1.6± 0.02  | 6.6± 0.3  | 1.7± 0.03  | 0.75± 0.01  | 1.9 ± 0.01 |
> > > | Full Bayes |                        |    0.37± 0.01 | 0.46± 0.01 |   |       |  |
> > >
> > > **Standard deviation:**
> > > | | Boston  |  Yacht  | Auto MPG | Wine  | Concrete  | Kin8nm | Facebook2 |
> > > |---|---|---|---|---|---|---|---|
> > > | Ours   | 0.3 ± 0.03 | 0.46 ± 0.08 | 0.36 ± 0.004 | 1.2 ± 0.03 |  0.63 ± 0.09 | 0.14 ± 0.003 | 0.17 ± 0.02 |
> > > | Vanilla GP | 0.79± 0.02  |   0.45± 0.005   | 0.4± 0.002  | 1.7± 0.05  | 0.49± 0.005  | 0.23± 0.005  |   |
> > > | Full Bayes |                        |   0.1± 0.001 |  0.12± 0.009 |   |       |  |

---

> > > > ### Comment · Reviewer_XbYQ · 2023-08-20
> > > >
> > > > Thanks for putting this up; apologies if this has already been mentioned + for the last minute comment but why are there no results for most of the datasets for the fully Bayesian GP?

---

> > > > > ### Author Response · Authors · 2023-08-20
> > > > >
> > > > > Thank you for your response. The fully Bayesian GP relies on a Markov Chain Monte Carlo method that scales very poorly with the amount of data. It becomes very computationally expensive after a couple hundred data points, making several repetitions prohibitive. Hence, we restrict ourselves to the two smallest datasets when applying the fully Bayesian approach. This is also the case for the results presented in the paper. To avoid any misunderstandings, this will be clearly stated in the revision whenever the fully Bayesian results are presented.

---

### Official Review · Reviewer_XLKC · 2023-07-06

**Soundness:** 3 good
**Presentation:** 3 good
**Contribution:** 3 good
**Rating:** 6
**Confidence:** 3

**Summary:**

This paper addresses the issue that the posterior variance of Gaussian processes are often poorly calibrated, typically underestimating quantile estimation. They propose a new method to calibrated uncertainty bounds, by training a quantity related to the posterior variance with new hyper parameters. This method further leverages optimization of the distance between the quantiles and the predictive mean, enabling sharp calibration. The method is tested on synthetic toy examples and standard UCI benchmark datasets, and appears to perform well against existing methods.

**Strengths:**

The paper is well written and presented and addresses a relevant problem, offering a principled solution.

**Weaknesses:**

The definition of a Gaussian process should specify that any **finite** collection of points are jointly Gaussian.

The most concerning issue is that the paper appears to be dismissive of two recent competing methods (Song et al.; Kuleshov & Deshpande) in the introduction, and does not mention them again. Indeed, the current work has advantages over each of these methods in terms of its construction and theoretical guarantees, however comparisons to these methods in terms of performance, as well as some further discussion, should help demonstrate the benefits of the proposed method. For example, the current method still relies upon a reasonably large dataset size, so dismissing a competing method because it only has asymptotic guarantees is maybe premature.

I would also question the presentation of the numerical results, in particular the use of both highlighting and boldface to denote best performing model in the tables. Since confidence intervals are provided, it may be advisable to bold the best performing ones, and using green to denote statistical significance that a single model is the best performing (i.e. multiple models can be bolded but none or one should be green).

**Questions:**

How does the proposed method compare with those proposed in Song et al, Kuleshov & Deshpande, as well as Capone, Lederer & Hirche?


**Limitations:**

 There is little view for potential negative societal impact due to improving the calibration of uncertainty bounds for Gaussian processes.

---

> ### Author Rebuttal · Authors · 2023-08-08
>
> Thank you very much for your review. We have made changes with your comments in mind as described below. If you feel that the changes sufficiently address your comments, we would be very thankful if you would consider raising your score.
>
> **Additional comparisons.** As you suggested, we compared our approach to that of Song et al. (2019), Capone et al. (2022) and Kuleshov and Deshpande (2022). We have included the results in the supplementary PDF. Note that the method of Capone et al. (2022) only computes uniform error bounds, i.e., 100 percent credible intervals, so we only considered this setting in the comparison.
>
> We noticed that training takes longer with the approaches of Kuleshov and Deshpande (2022) and Song et al. (2019) than our approach. This is because the optimization problem in our approach is fairly straightforward to solve, whereas Kuleshov and Deshpande (2022) and Song et al. (2019) require several optimization steps. Moreover, the approach of Song et al. (2019) is fairly involved and more difficult to implement than ours.
>
> We implemented the method of Kuleshov and Deshpande (2022) with the same neural network architecture suggested in their paper, trained over 2000 epochs. Our results suggest that the method of Kuleshov and Deshpande (2022) favors sharpness over calibration. While it achieves sharper intervals, it yielded significantly worse expected calibration errors than ours for most settings. Note that this is partially to be expected since their approach aims to achieve calibration in distribution, which only corresponds to quantile calibration if it is exact. Furthermore, it does not explicitly enforce calibration during training. We also noticed that it does not return valid intervals in some pathological cases (e.g., due to insufficient training steps).
>
> The method of Song et al. (2019) seemingly performs similarly to our approach in calibration, whereas it performed better in sharpness in some cases. However, we could only implement it on small data sets due to slow training. Furthermore, their method is fairly involved, and we relied heavily on the code kindly provided by the authors. Furthermore, although we employed the same test and training splits for the data as in our case, we noticed differences in the GP model used in their starting setup, potentially leading to significantly different starting models. We will ensure an identical setup for the revision.
>
> The method of Capone et al. (2022) is purely Bayesian and thus heavily dependent on the prior. We employed their approach as is from the code available online. The resulting credible intervals are well-calibrated, consistent with the results presented in their paper. However, our approach is much better regarding sharpness. This is because Capone et al. (2022) require symmetric intervals ($\text{mean} \pm b \times \text{standard deviation}$), whereas our approach allows for asymmetric credible intervals. Furthermore, our method does not rely on the prior distribution.
>
> **Presentation of numerical results.** As you suggested, we also removed the bolding everywhere except when methods perform significantly better, i.e., whenever the statistics indicate it is indeed best. In the revised version, we present the mean plus minus standard error and clearly state this to avoid confusion.

---

> > ### Comment · Reviewer_XLKC · 2023-08-18
> > **Response**
> >
> > I have read your response, as well as the responses to the other reviews. I am happy to raise my score in view of this, however note that there appear to be a large number of changes that may make others uncomfortable accepting this paper without seeing a revised version of the document.

---

> > > ### Author Response · Authors · 2023-08-21
> > >
> > > Thank you kindly for engaging with us and for taking the time to reassess our paper.
> > >
> > > We agree that the paper should not change significantly compared to the initial submission. In the present case, we are confident that the final paper will be largely identical to the initially submitted paper: the most significant changes pertain exclusively to the last section of the paper (Section 7), and have been presented in the PDF with the rebuttal. The only changes made to the main paper outside of Section 7 are the discussions of Assumption 4.2 and interpolant in Eq. (6), and the inclusion of the measurement noise in the vanilla GP prediction. However, these changes are very small and do not involve changing the core method in any way.

---

### Official Review · Reviewer_jTuD · 2023-07-17

**Soundness:** 3 good
**Presentation:** 3 good
**Contribution:** 3 good
**Rating:** 7
**Confidence:** 3

**Summary:**

The paper tackles calibration of Gaussian processes in regressions. The authors argue that while maximizing the evidence is a good way to choose hyperparameters to obtain an accurate posterior mean, it generally does not produces an accurate posterior variance. For this purpose, they propose a different way to obtain hyperparameters for the posterior variance which minimizes the length of centered confidence intervals, while constraining the parameters to provide accurate empirical quantiles. By replacing the constraint with a more treatable piece-wise expressions, they transform the problem into an unconstrained one. The further a practical algorithm and a theoretical result to obtain calibrated confidence intervals simultaneously over different confidence thresholds. Experimentation on toy and UCI datasets show the benefits of the method.

**Strengths:**

I think the paper is well written and clear. The solution proposed is reasonable and it tackles an important problem. The algorithms derived seem to be practical, and they are motivated by theoretical arguments that appear valid.

**Weaknesses:**

The authors bring up themselves that the core methodology they propose shares the same scalability issues as standard GPs, as it involves they inversion of a very large matrix. Hence, approximation methods are needed.

**Questions:**

- Not immediately clear to me what is the effect of replacing equation (5) with equation (6). I understand that the problem becomes easier to solve. Is the replacement, however, detrimental in some way? It is also not straightforward why the expression is (6) is the right one to use. I would encourage the authors to expand on this section.

- It is not super clear how easily the inequality in Assumption 4.1 holds. Perhaps making some examples, and discussing limitations of this assumption, would make this part clearer.


**Limitations:**

The work handles calibration for regression only.

---

> ### Author Rebuttal · Authors · 2023-08-08
>
> Thank you very much for your review and positive remarks. Based on your and other reviews, we have made several changes to improve the paper. Below you can find those that address your questions and comments specifically.
>
> Replacing (5) with (6) is not detrimental, as it still allows us to obtain a model that is sharp and calibrated. The choice of expression (6) is the simplest and most straightforward to compute. However, other forms of monotone interpolation are also possible without any loss of theoretical guarantees. In practice, the choice of interpolant eventually becomes of little relevance for high data sizes, since we only require small interpolation steps. We will include this discussion in the revised paper.
>
> Assumption 4.1 holds for any hyperparameter of a stationary kernel such that the Fourier transform of the kernel is monotonic with respect to that hyperparameter. This can be used to show, e.g., that the lengthscale of a stationary kernel satisfies Assumption 4.1 up to a scaling factor depending on the lengthscale (see Capone et al., (2022), Lemma 3.3). We will discuss this in more detail in the revision.

---

> > ### Comment · Reviewer_jTuD · 2023-08-17
> >
> > I thank the authors for their clarifications.
> >
> > After reading the concerns raised by the other reviewers, I agree with them that some experiments might have been clearer. However, I believe the authors made an effort to address all concerns and evaluate against additional competing methods.
> >
> > Personally, I do not see major reasons why this paper should be rejected, in particular after the further experimentation. The problem that this paper tries to address is clear to me, and the methodology is novel and well motivated by the theory. The main Theorem, in particular, provides a coverage guarantee with seemingly mild assumptions. The final algorithm is not particularly involved and fairly practical, apparently achieving good results.

---

> > > ### Author Response · Authors · 2023-08-21
> > >
> > > Thank you very much for engaging with us during the discussion phase.

---

### Author Rebuttal · Authors · 2023-08-09

We kindly thank all the reviewers for their helpful comments. They have helped significantly toward improving our paper. Below you will find a summary of the changes made to the paper. We have also attached a PDF with additional experiments and the revised toy problem.

**Presentation of numerical results.**
We have changed the presentation of the numerical results. In particular, we have removed the bolding whenever statistical evidence is insufficient to deem one method superior. We now present the numerical results as mean+- standard error, as opposed to mean+-standard deviation, as this better illustrates the expected deviation from the presented mean.

We have also modified the presentation of the toy example, as this caused some confusion. The toy example had been handpicked to illustrate the differences between methods concerning sharpness since all methods guarantee calibration. The new example presents calibrated results while clearly illustrating the differences between methods.

**Comparison to new methods.**
We have compared our approach to that of Song et al. (2019), Kuleshov and Deshpande (2022), and Capone et al. (2022). The results are presented in the PDF and detailed in the response to reviewer XLKC.

**Assumption 4.1.**
Assumption 4.1 can be shown to hold for kernels with Fourier transforms that are monotonically increasing with the hyperparameters. The revised paper shows how this can be leveraged to construct kernels that satisfy Assumption 4.1.

---

### Decision · Program_Chairs · 2023-09-21

**Decision:**

Accept (poster)

**Comment:**

This paper proposes a novel method for calibrating the predictive probabilities of a Gaussian process. The idea is to minimize the width of confidence intervals, while satisfying a calibration constraint on holdout data. The authors argue this produces sharper, yet calibrated estimates of uncertainty.

This paper addresses an important problem, and a new and useful idea is presented. Reviewers mostly complain about clarity, but the author feedback is helping a lot. While the experiments in the original submission have some limitations, new results posted during the rebuttal clarified concerns of reviewers. If this paper gets accepted, it is strongly recommended to improve the presentation of results (see reviewer comments), and to improve the overall clarity of presentation.